# Two distinct regulatory systems control pulcherrimin biosynthesis in *Bacillus subtilis*

**Nicolas L. Fernandez**[ID]**, Lyle A. Simmons**[ID] *

Department of Molecular, Cellular, and Developmental Biology, University of Michigan, Ann Arbor, Michigan, United States of America

* lasimm@umich.edu

## Abstract

Regulation of transcription is a fundamental process that allows bacteria to respond to external stimuli with appropriate timing and magnitude of response. In the soil bacterium *Bacillus subtilis*, transcriptional regulation is at the core of developmental processes needed for cell survival. Gene expression in cells transitioning from exponential phase to stationary phase is under the control of a group of transcription factors called transition state regulators (TSRs). TSRs influence numerous developmental processes including the decision between biofilm formation and motility, genetic competence, and sporulation, but the extent to which TSRs influence bacterial physiology remains to be fully elucidated. Here, we demonstrate two TSRs, ScoC and AbrB, along with the MarR-family transcription factor PchR negatively regulate production of the iron chelator pulcherrimin in *B. subtilis*. Genetic analysis of the relationship between the three transcription factors indicate that all are necessary to limit pulcherrimin production during exponential phase and influence the rate and total amount of pulcherrimin produced. Similarly, expression of the pulcherrimin biosynthesis gene *yvmC* was found to be under control of ScoC, AbrB, and PchR and correlated with the amount of pulcherrimin produced by each background. Lastly, our *in vitro* data indicate a weak direct role for ScoC in controlling pulcherrimin production along with AbrB and PchR. The layered regulation by two distinct regulatory systems underscores the important role for pulcherrimin in *B. subtilis* physiology.

## Author summary

Regulation of gene expression is important for survival in ever changing environments. In the soil bacterium *Bacillus subtilis*, key developmental processes are controlled by overlapping networks of transcription factors, some of which are termed transition state regulators (TSRs). Despite decades of research, the scope of how TSRs influence *B. subtilis* physiology is still unclear. We found that three transcription factors, two of which are TSRs, converge to inhibit production of the iron-chelator pulcherrimin. Only when all three are missing is pulcherrimin production elevated. Finally, we demonstrate that expression of pulcherrimin biosynthesis genes occurs via direct and indirect regulation by the trio of transcription factors. Due to its iron chelating ability, pulcherrimin has been

**Data Availability Statement:** All relevant data are within the paper and its Supporting Information files.

**Funding:** This work was supported by grant 1R35GM131772 from the National Institutes of

Health to LAS and by a Postdoctoral Research Fellowship in Biology award from the National Science Foundation (Award# 2010735) to NLF. The funders had no role in study design, data collection and analysis, decision to publish, or preparation of the manuscript.

**Competing interests:** The authors have declared that no competing interests exist.

characterized as a modulator of niche development with antioxidant properties. Thus, our findings that TSRs control pulcherrimin, concurrently with other developmental phenotypes, provides new insight into how TSRs impact *B. subtilis* and its interaction with the environment.

## Introduction

In the soil bacterium *Bacillus subtilis*, complex arrays of gene networks function together to precisely time the expression of gene products. A prime example is the series of decisions made as cells transition from exponential growth to stationary phase upon nutrient limitation [1]. This phase, termed the transition state, is where cells in the population use environmental cues to inform the next course of action to survive in the new environment, specifically whether to engage in competence, biofilm formation, motility, secondary metabolism, and/or acquisition of nutrients [1]. While there are many transcription factors controlling these processes, an important set among these are called transition state regulators (TSRs).

Originally defined in the context of sporulation, TSRs are regulators that inhibit expression of genes involved in developmental processes but do not result in a sporulation null mutation when deleted [2,3]. Notably, mutants in TSRs are still able to carry out post-exponential phenotypes, however the magnitude and timing of these phenotypes are disrupted [4]. ScoC and AbrB represent two well-studied TSRs in *B. subtilis*. ScoC is a MarR-family winged helix-turn-helix transcription factor first identified in hyper-protease mutants [5,6]. Microarray analysis between WT and *scoC* mutants identified 560 genes with differential gene expression involved in motility and genetic competence as well as protease production and peptide transport [7]. The smaller AbrB (10.8 kDa) is part of a large family of transcription factors with a beta-alpha-beta DNA binding N-terminal domain [8]. Mutants of *abrB* have pleiotropic effects and some regulatory overlap with ScoC [9,10]. ChIP-seq analyses of AbrB identified many binding sites with a bipartite TGGNA motif [9,11].

Pulcherrimin is a secreted iron chelating molecule that has been a topic of research in *B. subtilis* and other microorganisms [12–16]. Pulcherrimin is synthesized by first cyclization of two tRNA-charged leucines to form cyclo-L (leucine-leucine) (cLL), and second, oxidation to form water-soluble pulcherriminic acid (Fig 1A). Pulcherriminic acid is then transported outside of the cell where it can bind to free ferric iron to form the water-insoluble pulcherrimin (Fig 1A). While many microorganisms harbor the genes to produce pulcherrimin, the purpose of such a system to sequester iron is still unclear. Arnaouteli and coworkers found that pulcherrimin production contributed to growth arrest during *B. subtilis* biofilm formation through its ability to precipitate available extracellular iron [12]. Additionally, the ability of pulcherriminic acid to strongly sequester iron contributes to its anti-oxidative effects by limiting Fenton chemistry, providing evidence as an important antioxidant in cells [17,18].

Pulcherrimin biosynthesis is negatively regulated by the MarR-family transcription factor PchR, which is found in a cluster of two divergently transcribed gene-pairs encoding pulcherrimin biosynthesis (*yvmC* and *cypX*), regulation and transporter (*pchR* and *yvmA*) (Fig 1B) [12,13]. Interestingly, AbrB binding sites have been identified in the promoter for pulcherrimin biosynthesis and regulatory genes, suggesting AbrB was involved in pulcherrimin biosynthesis, however genetic analysis of AbrB regulation of pulcherrimin was not demonstrated [9,11]. Further, whether other TSRs are involved in the regulation of pulcherrimin production is not known.

**A)**

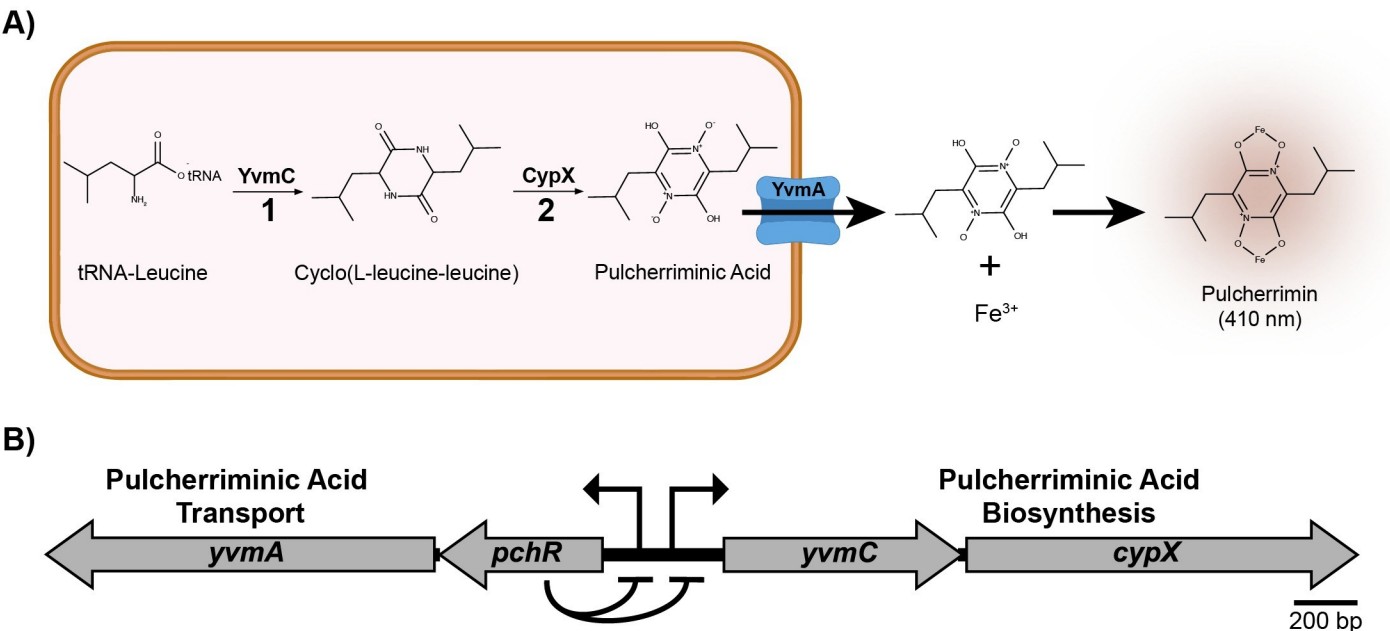

**B)**

**Fig 1. Pulcherrimin Biosynthesis, Transport, and Regulation in *B. subtilis*.** A) Pulcherriminic acid biosynthesis by the cyclization of tRNA-charged leucines to form cyclo(L-leucine-leucine) and the subsequent oxidation by CypX to form water-soluble pulcherriminic acid. Pulcherriminic acid is then transported out of the cell by YvmA, where it can form the insoluble pulcherrimin complex with iron, which forms a red color and has a peak absorbance at 410 nm. **B)** Genetic architecture of the pulcherrimin cassette. PchR, the MarR-family regulator encoded within the cassette, negatively regulates two promoters controlling *pchR-yvmA* and *yvmC-cypX* expression [13].

In this study, we provide evidence for a multi-layered regulation of pulcherrimin biosynthesis by the TSRs ScoC and AbrB as well as the pulcherrimin regulator PchR. We explore the kinetics of pulcherrimin production throughout the transition state and found that ScoC, AbrB, and PchR control the timing, rate, and amount of pulcherrimin produced by modulating expression of the pulcherrimin biosynthetic gene cluster *yvmC-cypX*. We further establish the roles of PchR and AbrB in direct regulation of gene expression utilizing *in vitro* DNA binding assays and provide evidence that ScoC can bind directly to the *yvmC* promoter *in vitro*. Together our results establish a model where pulcherrimin biosynthesis is regulated by nutrient levels during the transition from exponential phase to stationary phase in addition to input from PchR, linking stationary phase with extracellular iron sequestration.

## Methods

### Bacterial strains and culturing

A derivative of the wild-type strain 3610 *B. subtilis* harboring an amino acid substitution in the competence inhibitor ComI (Q12I) was used as the background strain in these studies [19]. Gene replacements and deletions were constructed as described [20]. Gel purified gene-targeting antibiotic resistance cassettes and non-replicative plasmids (see Cloning for construction details) were used to transform *B. subtilis* by natural transformation. Briefly, single colonies of the strain of interest were used to inoculate 1 mL LB supplemented with 3 mM $MgSO_4$ and grown to mid-exponential phase while shaking at 230 RPM at 37˚C. The cultures were then back diluted 1:50 into 2 mL MD media (1X PC buffer [10X PC—10.7 g $K_2HPO_4$, 6 g $KH_2PO_4$, 1.18 g trisodium citrate dehydrate, deionized water to 100 ml, filter sterilize], 2% glucose, 0.05 mg/mL tryptophan, 0.05 mg/mL phenylalanine, 0.01 mg/mL ferric ammonium citrate, 2.5

mg/ml potassium aspartate, 3 mM $MgSO_4$, water up to 2 mL) and grown for 3–5 hours, until early stationary phase. 10 μL of purified gene-targeting antibiotic resistance cassettes (~200–400 ng total) were added to 0.2 mL competent *B. subtilis*, were further incubated one hour, and plated on LB agar plates supplemented with either erythromycin (50 μg/mL), kanamycin (10 μg/mL), chloramphenicol (5 μg/mL), and/or spectinomycin (100 μg/mL). Antibiotic resistance clones were restruck on selection and insertions were verified by colony PCR using the US forward and DS reverse primers (See S1 Table). To remove the antibiotic resistance cassette, plasmid pDR224 was used to transform the appropriate strain with transformants selected for on LB supplemented with spectinomycin (100 μg/mL). Spectinomycin resistant clones were struck out on LB and incubated at the non-permissive temperature of 42˚C; this process was repeated twice. Clones were then rescreened for sensitivity of spectinomycin and the absence of the integrated antibiotic resistance cassette using PCR.

## Cloning

To generate gene disruptions, oligos were designed to amplify upstream (US) and downstream (DS) of the gene of interest with appropriate overhangs to fuse either an erythromycin or kanamycin resistance cassette (AbR) flanked by CRE recombinase recognition sites [20]. Oligonucleotides were designed using NEBuilder (NEB) with the default parameters except minimum overlap length was changed from 20 nucleotides to 30 nucleotides. Q5 polymerase (NEB) was used to amplify the appropriate PCR amplicon. All amplicons were gel extracted prior to assembly reactions (Qiagen). US, DS, and AbR fragments were assembled by splice by overlap extension (SOE) PCR (adapted from [21]). First, 0.5 pmol each of the US, DS, and AbrB amplicons were mixed with 18 μL Q5 5X buffer, 0.25 mM dNTPs, and water up to 89 μL. 1 μL Q5 (2U) was added, and PCR was carried out with the following parameters: 1 cycle of 98˚C– 10s, 10 cycles of 98˚C– 10s, 55˚C– 30s, 72˚C– 2 minutes, 1 cycle of 72˚C– 10 minutes. After completion of the PCR, 5 μL of US forward prime and 5 μL of the DS reverse primer were added and PCR was set under the following conditions: 1 cycle of 98˚C– 2 minutes, 15 cycles of 98˚C– 10s, 55˚C– 30s, 72˚C– 3 minutes, 1 cycle of 72˚C– 10 minutes. Following PCR, spliced amplicons were analyzed on a gel and 10 μL used directly for transformation of competent *B. subtilis*.

Vectors for protein purification (pNF039, pNF040, and pTMN007) and homologous recombination (pNF038) in *B. subtilis* were constructed using Gibson Assembly (NEB) following the manufacturer's protocol. Protein expression vectors were used to transform *E. coli* DH5alpha and homologous recombination vectors were used to transform *E. coli* MC1061 and clones were verified via sanger sequencing (Azenta) or whole plasmid sequencing (Eurofins). All primers and assembly methods are included in S1 Table. To generate a *pyvmC*-GFP transcriptional fusion (pNF047), plasmid pYFP-STAR was amplified with the primer pair oNLF554-oNLF555 and gel extracted. The ORF for sfGFP was amplified from plasmid pDR110-GFP(Sp) using the primer pair oLVG035A-oLVG035B and gel extracted. The *pyvmC* locus was amplified in two fragments: 1) oNLF524-525 and 2) oNLF526-527. The resulting fragments were then assembled by Gibson Assembly (NEB), and used to transform MC1061 *E. coli* cells by heat shock. Transformants were then selected for by plating on LB Amp plates. The assembled plasmid consisted of the *pyvmC* promoter lacking 17 nucleotides upstream of the ATG start codon of *yvmC*, deleting the native ribosome binding site which can contribute to spurious translation and high GFP background [22].

## Media

WT and derived *B. subtilis* strains were either grown in lysogeny broth (LB) or Tris-Spizizen salts (TSS) [Reagents added in order: 50 mM Tris pH 7.5, 136 μM trisodium citrate dihydrate,

water up to final volume, 2.5 mM dibasic potassium phosphate, 811 μM MgSO$_4$, 1X FeCl$_3$ from a 100X stock solution [150 μM FeCl$_3$, 0.1 g trisodium citrate dihydrate, 100 mL deionized water, filter sterilized], 0.5% glucose [25% stock solution, filter sterilized], and 0.2% ammonium chloride [20% stock solution, filter sterilized]. For liquid TSS, all components were mixed, filter sterilized, stored in the dark, and used within a week. For TSS agar plates, all reagents, except the 1X FeCl$_3$ solution, glucose, and ammonium chloride, were mixed with agar at 1.5% w/v and autoclaved. After the agar solution cooled to approximately 55°C, filter sterilized FeCl$_3$, glucose, and ammonium chloride were added, approximately 20–25 mL were added to sterile petri plates, and plates were allowed to dry overnight. TSS agar plates were stored at 4°C and were used within 6 months.

## Spot plating and liquid culture imaging

One day prior to the spotting, TSS plates with varying concentrations of FeCl$_3$ (final concentrations: 0.15, 1.5, 15, and 150 μM FeCl$_3$) were poured and dried overnight at room temperature. Spots for Fig 1 were on TSS plates supplemented with 150 μM FeCl$_3$. Strains were inoculated from frozen stocks into 1 mL TSS media and grown overnight at 37°C while shaking at 250 RPM. The next day, the turbidity of each culture was measured and adjusted to an OD$_{600}$ of 1.0 in fresh TSS media. 10 μL of each culture were spotted 15 mm apart on the same TSS plate and incubated for 24hours at 30°C. The next day, plates were imaged using an imaging box [23] and an iPhone 7 running iOS 15.7.5. Images were cropped and arranged using Adobe Photoshop and Illustrator. Each experiment included two technical spotting replicates and was repeated at least twice on separate days. For liquid cultures, 2 mL overnights were started from frozen stocks in TSS media and grown overnight at 37°C while shaking at 250 RPM. Overnight cultures were diluted to a starting OD$_{600}$ of 0.05 in 40 mL TSS in 125 mL flasks and grown for 20–24 hours at 37°C while shaking at 250 RPM. Images of the flasks were taken as stated above.

## LC/MS for cyclo-dileucine measurement

WT and *yvmC::erm* were struck out onto TSS plates from frozen stocks and grown overnight at 37°C for 16 hours. The next day, the strains were washed from the plate into fresh TSS and the resulting culture was used to inoculate 16 mL TSS in 50 mL flask at a starting OD$_{600}$ of 0.050. Cultures were grown for 6 hours while shaking at 250 RPM at 37°C. After six hours (OD$_{600}$ ~ 1.0), 15 mL of culture was collected by centrifugation in a 15-mL falcon tube (3 minutes, 4200xg) and the resulting pellets were resuspended in 200 μL cold extraction buffer (acetonitrile:methanol:water, 40:40:20, [24]). The resulting cellular mixture was centrifuged in a microcentrifuge (30s, 15,000 x g) and the supernatant containing the extracted metabolites were moved to a new 1.5 mL microcentrifuge tube and frozen at -80C. Samples were sent to the Michigan State University Research Technology Support Facility for LC-MS analysis of cyclo-dileucine. Two hundred microliters of the extract were evaporated to dryness using a SpeedVac and resuspended with an equal volume of methanol: water, 1:9 (v/v). Ten microliters of the sample was injected onto an Acquity Premier HSS T3 column (1.8 μm, 2.1 x 100 mm, Waters, Milford, MA) and separated using a 10 min gradient as follows: 0 to 1 min were 100% mobile phase A (0.1% formic acid in water) and 0% mobile phase B (acetonitrile); linear ramp to 99% B at 6 min, hold at 99% B until 8 min, return to 0% B at 8.01 min and hold at 0% B until 10 min. The column was held at 40°C and the flow rate was 0.3 mL/min. The mass spectrometer (Xevo G2-XS QToF, Waters, Milford, MA) was equipped with an electrospray ionization source and operated in positive-ion and sensitivity mode. Source parameters were as follows: capillary voltage 3000 V, cone voltage 30V, desolvation temperature 350°C, source

temperature 100˚C, cone gas flow 40 L/hr, and desolvation gas flow 600 L/Hr. Mass spectrum acquisition was performed in positive ion mode with a range of m/z 50 to 1500 with the target enhancement option tuned for m/z 227. A calibration curve was made using cyclo-dileucine standard (Santa Cruz Biotechnology). The peak area for cyclo-dileucine was integrated based on the extracted ion chromatogram of m/z 227.18 with an absolute window of 0.05 Da. Peak processing was performed using the Targetlynx tool in the Waters Masslynx software.

## Pulcherrimin isolation and measurement

Strains of interest were struck out on TSS agar plates from frozen stocks and incubated overnight (16–20 hours) at 37˚C. The next day, cells were collected by adding 1 mL fresh TSS media to the plates and gently swirled to remove bacteria adhered to the agar. The resulting bacterial suspension was then moved to microcentrifuge tubes, the $OD_{600}$ recorded, and diluted to a starting $OD_{600}$ of 0.05 in 40 mL TSS media in 125 mL round bottom flasks. At the time indicated, 1.5 mL of culture was aliquoted into a microcentrifuge tube. 0.1 mL were used for $OD_{600}$ measurement while the remaining cells were collected by centrifugation (10,000 x g, 30s). The supernatant was removed and the cell pellet and insoluble pulcherrimin were resuspended in 0.1 mL 2M NaOH by pipetting the solution until completely resuspended. The samples were then centrifuged again (10,000 x g, 1 minute) and the supernatant were moved to clean wells in a 96-well plate and absorbance at 410 nm was measured using a Tecan M200 plate reader.

To determine the time of entry into stationary phase, the R package growthrates was used to fit a linear growth model for every strain and replicate [25]. The time in which the growth data deviated from exponential growth was used as time 0 for Fig 2. The grow_gompertz3 function was used to model the change in absorbance at 410 nm over time using the growthrates package [25]. Each strain and replicate (n = 3) were modeled individually and the resulting parameters (maximum growth rate [mumax] and carrying capacity (abs. 410 nm) [K])

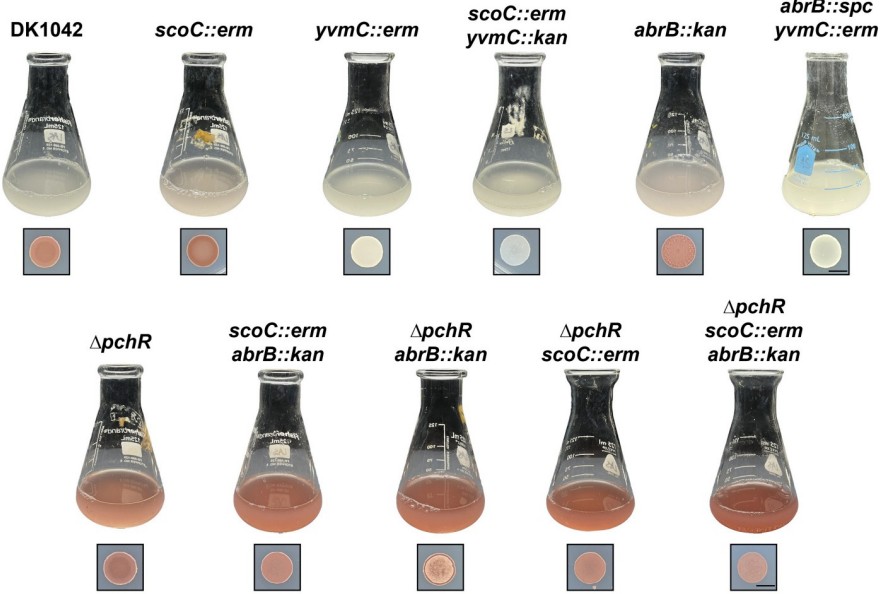

**Fig 2. Pulcherrimin production in liquid (top) and solid (bottom) TSS media.** WT (DK1042) and isogenic mutants were grown in liquid TSS media or spotted (10 μL) onto solid TSS media and grown overnight at 30˚C. The black scale marker corresponds to 5 mm.

were summarized by taking the average and standard deviation plotted in Fig 1B. For the production start time, curves were analyzed manually to determine when the predicted $A_{410}$ from the mutant strain deviated from the predicted $A_{410}$ value from the WT background during exponential phase. For the duration of pulcherrimin production, the x-axis distance between the beginning of the exponential phase of pulcherrimin production and the start of the stationary phase of pulcherrimin production were measured manually.

## Fluorescence reporter assay

WT and isogenic mutants harboring the *pyvmC*-GFP transcriptional fusion at the *amyE* locus were struck out on TSS agar plates and grown overnight for ~ 16 hours at 37˚C. The next day, the strains were plate washed into 1 mL TSS media and the $OD_{600}$ was recorded. 40 mL TSS in 125 mL Erlenmeyer flasks were inoculated with the plate washed cells at a starting $OD_{600}$ of 0.050. The cultures were incubated at 37˚C while shaking at 250 RPM until the cultures reached mid-exponential phase. Fluorescence was measured from a 1 mL sample using an Attune NxT Acoustic Focusing Cytometer (ThermoFisher Scientific) using the following settings: Flow rate, 25 µl/min; FSC voltage, 200; SSC voltage, 250; BL1 voltage, 250 [26]. A WT strain not harboring GFP was used as a control to assess background fluorescence (grey bars in Fig 3A). Percent GFP positive was calculated by dividing the number of fluorescent events with a signal above the maximum fluorescence of the negative control (No GFP) by the total number of fluorescent events of a given strain and multiplying by 100. This was repeated for each replicate for each strain and the mean +/- standard deviation of the percent positive values are provide to the right of each panel in Fig 3A.

## Protein purification

ScoC purification was carried out as previously described with some minor amendments [26]. Plasmid pTMN007 harboring ScoC in a T7 expression vector pE-SUMO was used to transform BL21-DE3 *E. coli* and plated on LB supplemented with kanamycin (25 µg/mL). The next day, a single colony was inoculated into 1 mL LB $Kan_{25}$, grown to mid-exponential phase at 37˚C while shaking at 230 RPM, diluted 1:5 in 5 mL LB $Kan_{25}$, and grown overnight at 37˚C while shaking. After overnight growth, the culture was diluted to a starting $OD_{600}$ of 0.05 in 400 mL LB $Kan_{25}$ at 37˚C shaking at 230 RPM and grown until the $OD_{600}$ reached between 0.7, after which 1 mM final concentration of Isopropyl ß-D-1-thiogalactopyranoside (IPTG) was added to induce protein expression for three hours at 37˚C. After induction, cells were collected by centrifugation (5 minutes at 7,500 RPM using SLA-1500 rotor in a Sorvall RC 5B plus centrifuge) and cell pellets were stored at -20˚C until use. Cell pellets were thawed at room temperature and resuspended in 40 mL lysis buffer (50 mM Tris pH 8, 300 mM NaCl, 10% sucrose, 10 mM imidazole, and 1 EDTA-free protease inhibitor tablet added the day of purification (Roche)). The cell solution was moved to a beaker in an ice-water bath and sonicated (15s ON, 25s OFF, 24 cycles, 50% amplitude, Fisher Scientific Model 505 Sonic Dismembrator). The lysate was cleared by centrifugation (10 minutes, 12,000 RPM, using an SS-34 rotor in a Sorvall RC 5B plus centrifuge). The clarified lysate was loaded onto a 3 mL Ni-NTA column pre-equilibrated with lysis buffer and the flow through was discarded. The column was washed three times with 20 mL wash buffer (50 mM Tris pH 8, 2M NaCl, 25 mM imidazole, and 5% glycerol) and ScoC-SUMO-His was eluted with 15 mL elution buffer (50 mM Tris pH 8, 150 mM NaCl, 200 mM imidazole). The protein solution was dialyzed at 4˚C into dialysis buffer (50 mM Tris pH 8, 150 mM NaCl, 5% glycerol). The next day, DTT (1 mM) and SUMO Ulp1 protease were added, the solution was incubated at room temperature for 2 hours and dialyzed into dialysis buffer overnight at 4˚C. Following dialysis, SUMO-free ScoC was purified by loading the

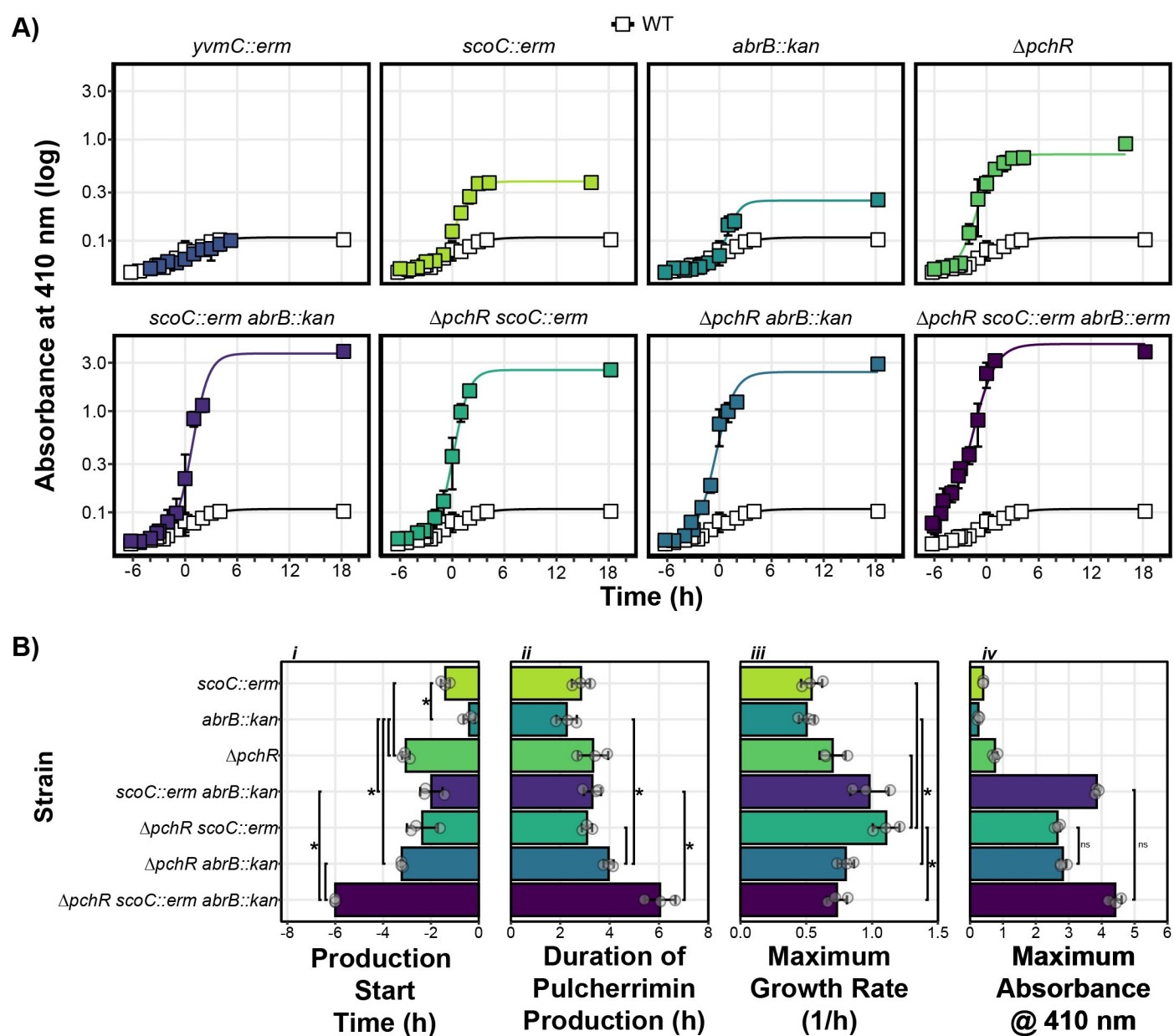

**Fig 3. ScoC, ArbB, and PchR Control Timing and Rate of Pulcherrimin Production. A)** Pulcherrimin production was measured as a function of growth phase, where T0 marks the transition from the exponential growth to stationary phase. Each panel represents the average $A_{410}$ for a given strain compared to the $A_{410}$ from WT (white squares). Error bars represent +/- the standard deviation. Lines running through the points are modeled using the *drm* function from the *drc* package in R (see Methods and Materials). **B)** Pulcherrimin production parameters as a function of genetic background: *i)* start of pulcherrimin production time relative to the transition phase of growth (T0), *ii)* the duration of pulcherrimin production, *iii)* the maximum estimated production rate, and *iv)* the maximum absorbance at 410 nm. For panels *i-iii*, brackets and asterisks indicate significant comparisons. For panel *iv*, brackets and "ns" indicate non-significant comparisons, where every other comparison had an adjusted p-value less than 0.05 as determined by T-test corrected for multiple comparisons with the Bonferroni correction.

solution onto 3 mL of pre-equilibrated Ni-NTA resin by collecting the flow through. SUMO-free ScoC fractions were determined via SDS-PAGE, pooled, quantified by the Bradford assay, diluted with glycerol for a final concentration of 25%, and stored at -80˚C.

Expression vectors for PchR and ArbB were constructed similar to ScoC. Growth of BL21 *E. coli* harboring PchR was identical to ScoC. Growth of BL21 *E. coli* harboring ArbB had the following changes. First, 1 mL LB Kan was inoculated with a single colony of *E. coli* harboring

the AbrB expression vector and grown for 6 hours at 37°C, shaking at 200 RPM. The 1 mL culture was diluted 1:10 in 9 mL LB Kan in a 125 mL flask and grown overnight at 37°C while shaking at 200 RPM. The next day, the culture was used to inoculate 400 mL of LB Kan at a starting $OD_{600}$ of 0.1 and grown at 30°C until the $OD_{600}$ reached between 0.6 and 0.7, at which point IPTG was added at a final concentration of 1 mM and the culture was moved to 16°C with shaking at 160 RPM for 16 hours. For purification, slight modifications to the lysis buffer and elution buffers were made. Frozen pellets of PchR-SUMO and AbrB-SUMO were resuspended in 40 mL lysis buffer (50 mM Tris pH 8, 500 mM NaCl, 10% glycerol, 20 mM imidazole, supplemented with 1 EDTA-free protease inhibitor tablet), lysed by sonication, and the lysate cleared by centrifugation. Clarified lysate was applied to 3 mL Ni-NTA resin columns, the columns were washed with 60 mL lysis buffer, and proteins were eluted by step elution using 5 mL each of increasing imidazole concentrations (elution buffer: 50 mM Tris pH 8, 500 mM NaCl, 10% glycerol, imidazole at 50, 100, 200, and 350 mM). Elution fractions were assayed for relative protein concentration by the Bradford assay (BioRad) and fractions containing protein were electrophoresed on SDS-PAGE to ensure proper expression and purification. Removal of the SUMO tag and purification of SUMO-free protein was carried out as described above. SUMO-free AbrB required an additional anion exchange purification step using a HiTrap qFF (Cytivia 17515601) anion exchange column attached to an AKTA FPLC. The column was equilibrated with 10% Q Buffer B (50 mM Tris, 5% glycerol, and 500 mM NaCl). Sumo-free AbrB was diluted to 50 mM NaCl in Q Buffer A and loaded into the column. Protein fractions (2 mL) were collected as the system increased the percentage of Q Buffer B while monitoring A260 readings. High A260 peaks were measured for AbrB on SDS-PAGE and correct fractions were pooled, dialyzed, concentrated by dialysis, mixed with glycerol at 25% final concentration, and stored at -80°C.

## Electrophoretic mobility shift assays

5′ IRDye 700-labeled probes of the *yvmC* promoter (-244 to +9 relative to the ATG start codon) were generated by PCR using the primer pair oNLF433-oNLF387 using pNF035 as a template. PCR products were purified by gel extraction and quantified by nanodrop. To generate the *PyvmCΔ59* probe, two PCR reactions were carried out with primer pairs oLVG025A-oNLF467 and oNLF468-oNLF387. The two PCR products were gel extracted and fused together by SOE PCR (see Cloning). The fragment was then used as a template for PCR with primer pairs oNLF433-oNLF387, resulting in a 5′ IRDye 700 labeled DNA fragment lacking 59-bp. Binding reactions were assembled by first generating a binding solution: 1X binding buffer (5X binding buffer: 250 mM Tris pH 8, 5 mM EDTA, 150 mM KCl, 10 mM $MgCl_2$, 12.5 mM DTT, 1.25% Tween 20, and 2.5 mg/mL BSA), 1X DNA probe (10X probe, 100 nM), 1 μL protein of interest (5X stock concentration), and water up to 5 μL. The binding reactions were incubated for 30 minutes at 37°C. When indicated, 1 μL of 1X heparin (6X heparin: 0.06 mg/mL) was added to each reaction after incubation and 3 μL were loaded into the wells of a 15-well 6% polyacrylamide gel and electrophoresed for 60 minutes at 150V at room temperature. After gel electrophoresis, the gels were left in the glass plates and imaged using an Odyssey xCl imager (1.5 mm offset height, 84 μm resolution). The resulting images were adjusted in Fiji [27] and cropped and annotated in Adobe Illustrator. EMSA experiments were carried out at least three times with separate aliquots for each protein.

## Fluorescent DNAse I and Differential peak height analysis

5′ FAM-labeled probes were generated by PCR using the primer pair oNLF432-oNLF387 using pNF035 as a template. PCR products were purified by gel extraction and quantified by

nanodrop. Binding reactions were assembled and carried out identically to EMSA experiments. After incubation, 1 μL 0.6 mg/mL heparin, and 0.79 μL of 10X DNAse I buffer (Invitrogen) were added to the binding reactions followed by 1.2 μL of diluted DNAse I (0.625 U total, Invitrogen) and reactions were incubated at room temperature for 5 minutes. After 5 minutes, reactions were heated to 72˚C for 10 minutes and DNA was immediately purified by phenol-chloroform extraction and ethanol precipitation. Reactions were resuspended in 10 μL MiliQ water and submitted for fragment analysis by Azenta. The resulting.fa files were imported into R and the data aligned to the LIZ500 reference standards using the *storing.inds* and *overview* functions from the R package Fragman [28]. To analyze differences between peak heights, the R function findPeaks were used to extract local maxima in the sequential peak height data [29]. Differential peak height analysis was carried out as described in [30]. Briefly, the raw signal for each sample was normalized by dividing by the sum of all signals. Then, the normalized signal for the sample incubated with protein was subtracted from the signal incubated without protein, producing the differential peak height. DNase I protection produces negative values while DNase I hypersensitivity produces positive values. DNase I footprinting and differential peak height analysis were carried out at least twice with separate protein aliquots.

## Results

### ScoC Negatively regulates pulcherrimin production

The TSR ScoC represses gene expression during exponential phase [5,10,31]. As nutrients become limiting, the effect of ScoC repression is lessened by downregulation of *scoC* expression and competition between other DNA binding proteins [10,31,32]. Our lab has identified ScoC as a methylation-responding transcription factor at a promoter of a gene not involved in protease production, thus we were interested in further characterizing the role of ScoC as a TSR in *B. subtilis* [26]. When culturing our WT strain (DK1042) to late stationary phase cultures appear grey in liquid minimal media (Tris-Spizicen Salts, TSS) and appear red on TSS agar (Fig 2). An isogenic *scoC* disruption mutant (*scoC*::erm), interestingly, appears pink in liquid TSS and has a more intense coloring when grown on solid TSS plates compared to WT (Fig 2). The red phenotype was dependent on the amount of $FeCl_3$, not present on standard LB plates, and present on LB plates supplemented with $FeCl_3$ (S1 Fig). Thus, excess iron was responsible for the red phenotype.

   *B. subtilis* and many other microorganisms produce and secrete the iron chelator pulcherriminic acid, which binds to free ferric iron to form the insoluble pigment pulcherrimin [14,15,33] (Fig 1A). We hypothesized that the red phenotype was caused by the production of pulcherrimin. Therefore, we made mutations in one of the two key pulcherrimin biosynthetic genes (*yvmC*) in the WT and *scoC::erm* backgrounds. After growth in liquid and on solid media, the red phenotype in the mutants lacking *yvmC* was absent, indicating pulcherrimin is responsible for the red phenotype in both the WT and *scoC::erm* backgrounds (Fig 2). Additionally, mass spectrometry analysis of cyclo(L-leucine-L-leucie) (cLL), a pulcherriminic acid precursor synthesized by the cyclization of tRNA-charged leucine by YvmC, showed that mutants lacking *yvmC* no longer have detectable cLL (S2 Fig). Further, complementing the *scoC::erm* by ectopically expressing *scoC* from its native promoter causes the red phenotype to disappear (S3 Fig). Together, our results indicate that ScoC negatively controls pulcherriminic acid production, in turn resulting in increased extracellular pulcherrimin and the red color in TSS media in the *scoC* mutant.

### Multiple systems control pulcherrimin Production in *B. subtilis*

Past studies have identified AbrB, another TSR, as a regulator of pulcherrimin production in *B. subtilis* and *Bacillus licheniformis* [9,15]. We therefore generated an *abrB* disruption strain

(*abrB::kan*) and assessed pigment formation after overnight growth. Like *scoC::erm*, the liquid media turned slightly pink relative to WT. On solid media, *abrB::kan* appeared wrinkly, was less red than *scoC::erm*, and more red than WT (Fig 2). Introducing a *yvmC* disruption in the *abrB* mutant background also abolished pigment formation, indicating pulcherrimin production is increased in the *abrB* mutant background (Fig 2). The wrinkly phenotype of the *abrB::kan* colony is due to its role in negatively regulating biofilm formation [34].

Many bacteria that encode the pulcherriminic acid biosynthesis genes also encode the negative regulator *pchR* which inhibits expression of the biosynthesis operon [12]. In *B. subtilis*, cells lacking *pchR* appear red in liquid culture after overnight growth in minimal media (MS media) [13]. Indeed, deletion of *pchR* (*ΔpchR*) resulted in a more intense red coloring of the liquid and solid TSS media compared to WT and the other single mutants (Fig 2). Combining the mutations in the same background, resulting in three double-mutants and one triple mutant, results in a color intensity higher than any single mutant alone (Fig 2). However, whether there are differences in the amount, or even the rate at which pulcherrimin is produced, is not easily determined with qualitative comparisons, necessitating a quantitative assessment of pulcherrimin production during different growth phases (see below).

## ScoC, AbrB, and PchR Control the Timing, Rate, and Amount of Pulcherrimin Produced in Liquid Cultures

Iron bound pulcherriminic acid (pulcherrimin) is water-insoluble at neutral pH and can be sedimented with cells through centrifugation and solubilized in a basic solution (2 mM NaOH). In this solution, pulcherrimin can be analyzed spectrophotometrically with peak absorbances at 245, 285, and 410 nm [16,35]. The amount of pulcherrimin produced in a culture growing over time can be determined by measuring the absorbance at 410 nm ($A_{410}$) from alkali-solubilized cell pellets. We were interested in how pulcherrimin production changed as cells transitioned from exponential growth to stationary phase. Therefore, we measured the absorbance at 410 nm ($A_{410}$) in the WT background and found that it rose steadily throughout the growth curve, plateauing at 0.10 after 18 hours post transition into stationary phase (Fig 3A). Interestingly, the $A_{410}$ in the *yvmC::erm* background, which lacks an enzyme necessary for the pulcherrimin precursor cLL, is indistinguishable from WT (Figs S2 and 3A). These data demonstrate pulcherrimin production in the WT background grown in liquid TSS culture is below the limit of detection for this assay and that the $A_{410}$ in WT and *yvmC::erm* represents the background absorbance.

We repeated this experiment with all strains and analyzed the data using a growth model to estimate the start time, the duration, the rate, and the maximum amount of pulcherrimin produced (Fig 3A and 3B). Concomitantly, we monitored bacterial growth by measuring absorbance at 600 nm and found similar growth trajectories for all strains (S4 Fig). Genes controlled by TSRs tend to have low expression during exponential phase and higher expression as cells transition into stationary phase [2,10,32,36–38]. We hypothesized that the effect of *scoC* and *abrB* disruption would cause increased pulcherrimin production as cells transition into stationary phase while cells without *pchR* would have higher pulcherrimin production during exponential phase. Compared to any single mutant, *ΔpchR* had the earliest production start time while *abrB::erm* began pulcherrimin production just prior to the start of the transition phase (Fig 3A and 3Bi). The maximum production rate and duration of pulcherrimin production were similar across all single mutants while *ΔpchR* had the highest maximum $A_{410}$ (Fig 3Bi–3Biv). The data from the single mutants suggests *pchR* is a potent repressor of pulcherrimin production, especially during exponential phase while *scoC* and *abrB* contribute to repress production during late exponential through early stationary phase.

Next we asked how combining mutations to generate double and triple mutants affected pulcherrimin production parameters. When comparing against the single mutants, combining *scoC::erm* and *abrB::kan* resulted in a production start time and production duration comparable to the single *scoC* disruption (Fig 3Bi–3Bii). However, the *scoC::erm abrB::kan* background had an increased pulcherrimin production rate and a maximum $A_{410}$ much higher than the single mutants (Fig 3Biii–3Biv). Introducing *scoC::erm* or *abrB::kan* into the *ΔpchR* background generated strains with production start times and pulcherrimin production durations similar to *ΔpchR* (Fig 3Bi). While the maximum production rate was higher in *ΔpchR scoC::erm* than *ΔpchR abrB::kan*, the maximum $A_{410}$ were similar (Fig 3Biii–3Biv). This can be reconciled by the fact that *ΔpchR abrB::kan* has a longer duration of pulcherrimin production than *ΔpchR scoC::erm* (Fig 3Bii). In the triple mutant background, pulcherrimin production began at the first sampled timepoint, six hours before the transition into stationary phase, demonstrating all transcription factors contribute to inhibiting expression during exponential phase (Fig 3A and 3Bi). The triple mutant had the longest duration of pulcherrimin production of all mutants tested while the maximum production rate was lower than most double mutants and similar to all single mutants. Despite the lower production rate, the maximum $A_{410}$ was similar to or higher than all double mutants, likely because of the extended production period (Fig 3Bii–3Biv). The fact that the double and triple mutants demonstrated synergistic gene interactions suggests ScoC, AbrB, and PchR act independently to regulate pulcherrimin production. We conclude that multiple transcription factors control the rate, duration, and maximum amount of pulcherrimin produced and demonstrate an integration of multiple regulatory systems on an energetically costly phenotype.

## The *yvmC* Promoter is Upregulated in the Absence of ScoC, AbrB, and PchR

As transcription factors, we hypothesized the main role of ScoC, AbrB, and PchR in controlling pulcherrimin formation likely involves regulating promoter activity. We therefore fused the promoter for *yvmC* ($P_{yvmC}$) to GFP and measured single-cell fluorescence via flow cytometry during early exponential phase (approximately 3 hours before T0). 69.8% (+/- 9.87%) of the WT population were GFP positive compared to a no GFP control (Fig 4A). The percent positive population of cells in the *scoC::erm* and *abrB::kan* backgrounds were greater than WT. Further, the wider distribution of fluorescence intensities indicate a broader range of expression levels within the population compared to WT. We found a proportional relationship between maximum $A_{410}$ values and reporter expression for most strains (Figs 3Biv and 4). Additionally, mutants with higher reporter activity tend to have more narrow fluorescence distributions, except for the double mutant *scoC::erm abrB::spec*, which has a wider fluorescence distribution (Fig 4). Together, our results demonstrate PchR, ScoC, and AbrB work to inhibit expression of the pulcherrimin biosynthesis genes *yvmC* and *cypX* and that increased promoter expression results in increased pulcherrimin production.

## Analysis of Transcription Factor Binding at the *yvmC* Promoter

Randazzo and coworkers aligned promoter regions of PchR regulated genes and identified a 14-bp consensus sequence termed the PchR-box [13]. To validate their *in silico* consensus motif, we utilized fluorescent DNase I footprinting assays followed by differential peak height analysis to identify the PchR binding site *in vitro*. PchR demonstrated a protected region from +6 to +28 relative to the transcriptional start site (determined by [39]), which overlaps with the previously identified PchR-box (Fig 5Ai–5Aii, [13]).

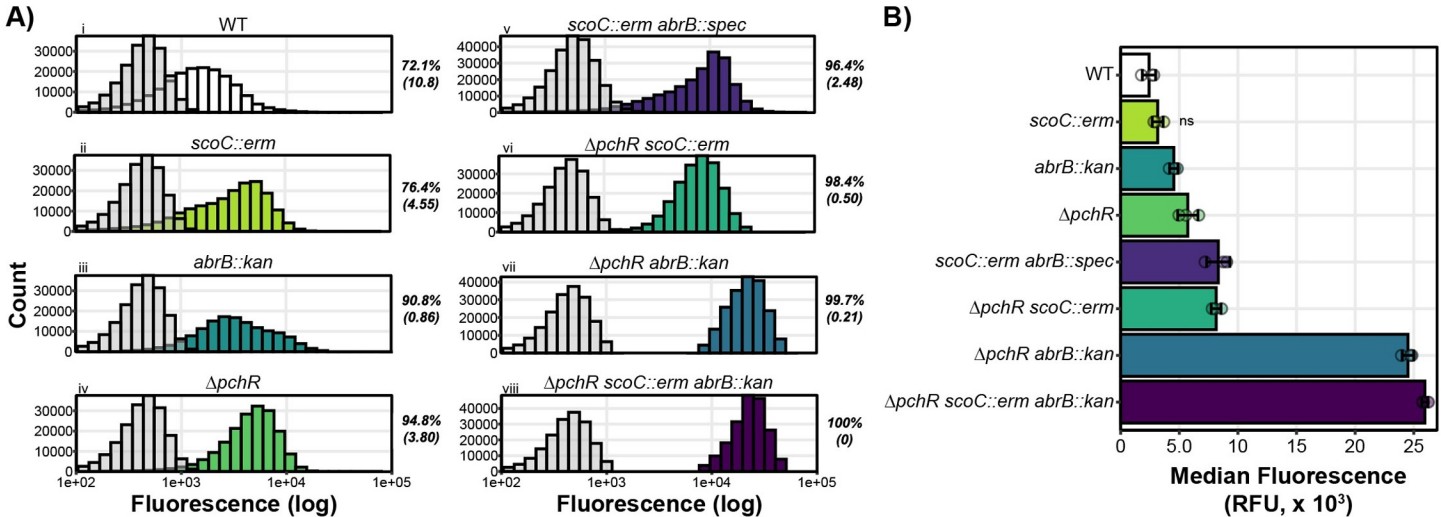

**Fig 4. ScoC, PchR, and AbrB Repress the *yvmC* Promoter. A)** Histograms representing fluorescence distribution as a function of cell count. In each panel, the negative control (grey, WT without GFP) was plotted with the corresponding genotype harboring a *yvmC* promoter fusion to GFP. Mean percent GFP positive with standard deviation in parentheses is provided to the right of each plot. Fluorescence was measured independently for each strain on three separate days with a representative shown. **B)** Median fluorescence for three separate trials (bar) with each trial median shown (circles). Error bars represent stand deviation between trials. Asterisks indicate an adjusted p-value less 0.05 while "ns" indicate non-significant comparisons with WT.

We found that AbrB had a broad protection area encapsulating -30 to approximately +60 bp relative to the transcriptional start site (Fig 5B). While broader than PchR, the protection region of AbrB is consistent with earlier reports of AbrB-DNA interactions [40]. Interestingly, attempts at DNase I footprinting analysis with ScoC and the *yvmC* promoter were unsuccessful, with no apparent difference in peak heights between samples with and without protein (S5 Fig). During optimization experiments, we found that ScoC could bind to the *yvmC* promoter when the non-specific competitor poly dI-dC was used, rather than the polyanionic compound heparin used in experiments with PchR and AbrB. We therefore analyzed the DNase I footprint with ScoC using poly dI-dC as a non-specific competitor and observed a broad protection area from approximately -10 to + 60 relative to the transcriptional start site (Fig 5C).

The footprint data indicated all proteins interacted with the *yvmC* promoter within the same region upstream and downstream of the transcriptional start site (Fig 5). We therefore generated a deletion probe where 59 base pairs, from -14 to +45 bp, were deleted (Δ59) and assessed DNA binding *in vitro* by electrophoretic mobility shift assays (EMSA) using the WT and Δ59 probes. PchR exhibited the most canonical behavior of the transcription factors, demonstrating discrete band formation as protein concentration increased (Fig 6A). At the highest concentration tested, PchR formed a second DNA-bound species that migrated slower than the other band formed at lower concentrations, suggesting an additional, low-affinity site may be present in the promoter (Fig 6A). However, DNase I footprinting analysis identified only one area of protection, suggesting the slow-migrating complex may be caused by non-specific interactions between PchR and DNA at the high protein concentration (Fig 5A). Indeed, when using the Δ59 probe, which lacks the region recognized by PchR, there is a faint shift at the highest PchR concentration (Fig 6B). This suggests that high PchR concentrations can interact with the *yvmC* promoter non-specifically but demonstrates specific binding at lower concentrations.

For AbrB, the intensity of the unbound probe decreased as protein concentration increased, indicating that DNA binding is occurring. However, the lack of discrete bands indicate the

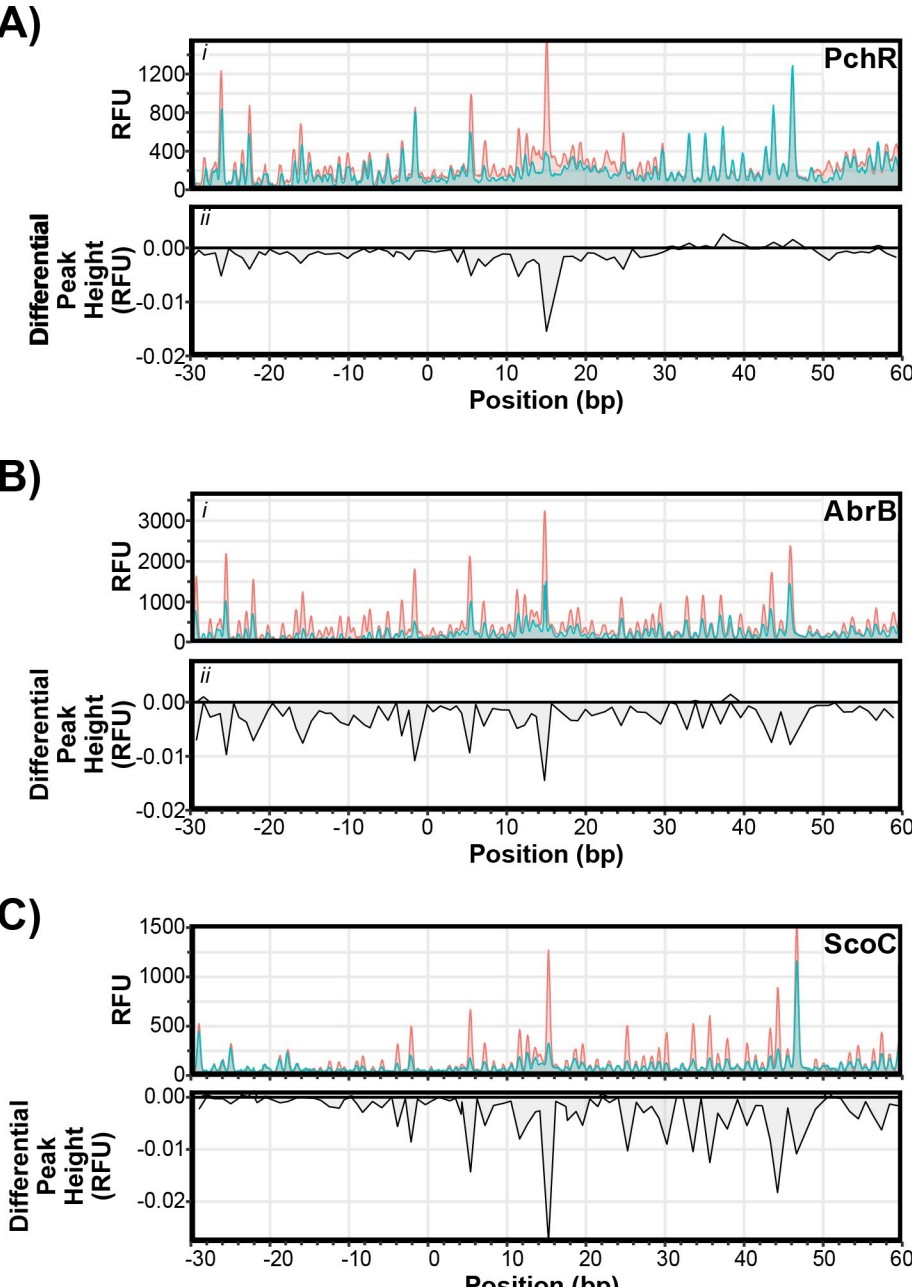

**Fig 5. DNase I Protection Varies Among ScoC, PchR, and AbrB.** Electropherograms of fluorescent DNase I footprinting analysis of PchR (A, 250 nM), AbrB (B, 500 nM), and ScoC (C, 1000 nM) with fluorescently labeled *yvmC* promoter as a function of estimated nucleotide position. Fluorescence intensity (RFU, panel *i*) of reactions incubated with (blue) and without protein (red). Differential peak height (panel *ii*) between reactions with protein and without protein. Differential peak heights less than zero indicate protection while differential peak heights greater than zero indicate hypersensitivity.

AbrB-*PyvmC* interaction likely represent a fast off-rate (Fig 6A). Additionally, protein binding was not observed in experiments using the Δ59 probe (Fig 6B). ScoC had smear shifts at concentrations greater than 250 nM with a band present at 1000 nM (Fig 6A). Interestingly, a band was also present at 1000 nM when using the Δ59 probe but not at 250 or 500 nM,

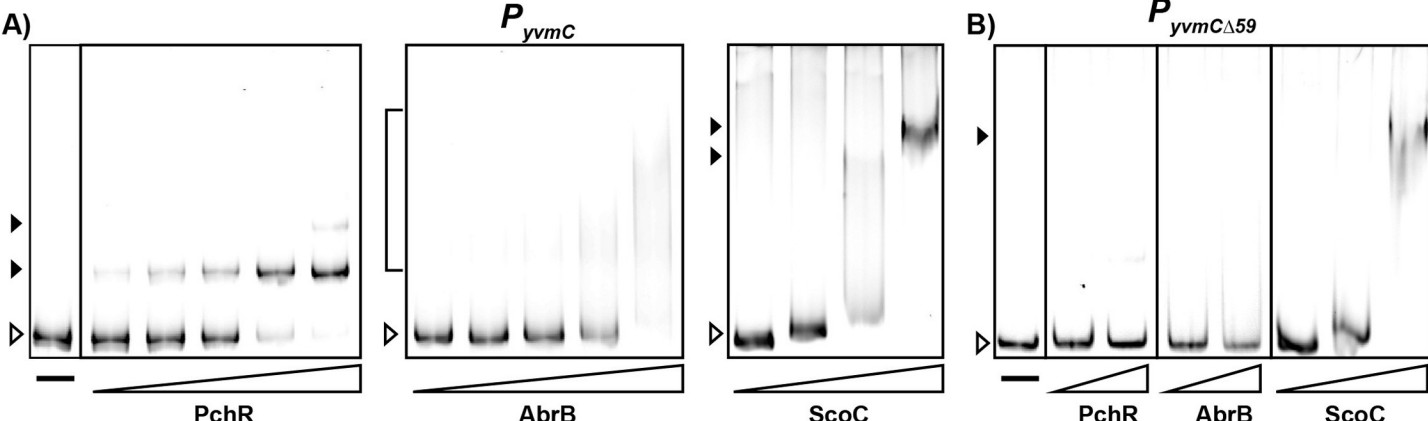

**Fig 6. ScoC, PchR, and AbrB Bind Near the Core Promoter Region of *yvmC*.** Electrophoretic mobility shift assays with WT $P_{yvmC}$ (A) and the Δ59 promoter $P_{yvmCΔ59}$ (B) with increasing concentrations of purified PchR (left panel), AbrB (middle panel), and ScoC (right panel). Unshifted bands are marked with unfilled triangles, shifted bands are marked with filled triangles, and smears are marked with brackets. Proteins were diluted two-fold and final concentrations are as follows: PchR (15.6 to 250 nM for WT, 125–250 nM for Δ59), AbrB (62.5 to 1000 nM for WT, 500–1000 nM for Δ59), and ScoC (125 to 1000 nM for WT, 250–1000 nM for Δ59).

suggesting binding at 1000 nM occurs non-specifically (Fig 6B). Previous studies of *in vitro* ScoC-DNA interactions found that ScoC binds to DNA non-specifically at concentrations greater than 400 nM and the DNase I footprint size is around 14–25 bps [31]. The fact that ScoC bound to the WT but not the Δ59 probe indicates specificity, but it is apparent that the ScoC-*yvmC* interaction likely has a fast off-rate as suggested with AbrB.

Nonetheless, the results suggest PchR and AbrB, collectively, act as road-blocks to RNAP progression at the *yvmC* promoter (Fig 7A and 7B). The role of ScoC involves weak direct regulation and may also have indirect regulation through another factor that ScoC modulates. Taken together, the presence of the three repressors is necessary to limit the production of pulcherrimin providing mechanistic insight into the regulatory network of pulcherrimin production in bacteria.

## Discussion

In this study, we uncover the layered negative regulation of the iron chelator pulcherrimin by two discrete regulatory systems. We demonstrate that these systems work together to inhibit the biosynthetic pathway of an energetically costly and potentially growth limiting metabolite. While expression patterns of *yvmC*-GFP promoter fusion closely resembled the pattern of pulcherrimin production, the reporter in the *scoC::erm* and *abrB::kan* background displayed broad distributions of promoter expression, suggesting the TSRs influence heterogeneity in pulcherrimin producing cells. Only in the absence of all three transcription factors is *yvmC* gene expression fully achieved and pulcherrimin is produced throughout exponential phase. Our biochemical analysis shows the regulator of the pulcherrimin biosynthesis operon and the transition state regulators ScoC and AbrB bind to the *yvmC* promoter with differing apparent off-rates. Together, our results indicate pulcherrimin regulation in *Bacillus subtilis* is under tight regulation and repressed during exponential growth by two distinct regulatory systems.

The transition state regulators ScoC and AbrB have been subject to much research for their roles in diverse aspects of *Bacillus* physiology. The absence of these regulators modifies sporulation, competence, protease production, and biofilm formation among many other phenotypes [7,10,36,38,41–44]. Microarray analysis of the effect of *scoC* on global gene expression was carried out from cells sampled at different points in the growth cycle in complex media

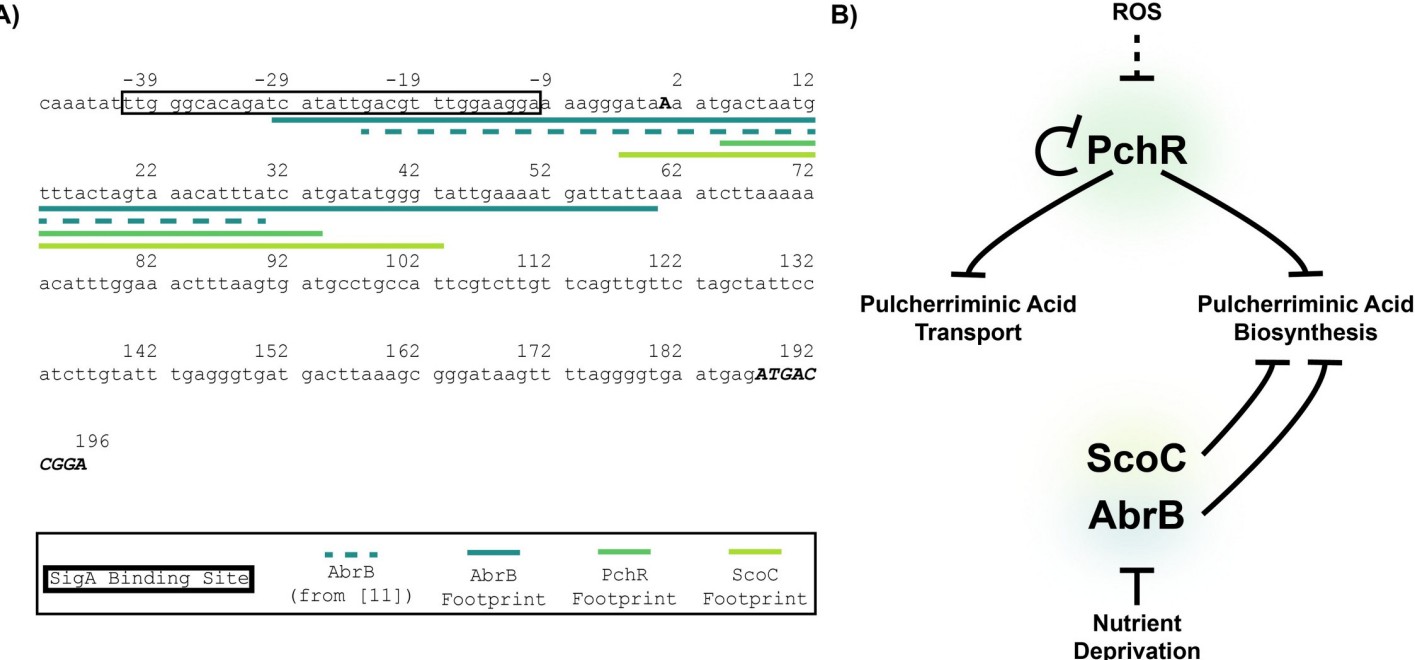

**Fig 7. Model of Pulcherrimin Regulation by ScoC, AbrB, and PchR. A)** Promoter sequence of *yvmC*. The putative SigA binding site is marked with a box and the transcriptional start site is bold and upper case. Regions protected by transcription factors in the DNase I footprinting assays are provided by colored lines under the DNA sequence. The dashed line represents the sequence in the region pulled down by AbrB *in vivo* by Chumsakul et al [11]. The translation start site and the first 9 base pairs of *yvmC* are bold, italicized, and in upper case. **B)** The MarR-family transcription factor PchR negatively regulates its own expression as well as the genes needed for pulcherriminic acid biosynthesis (*yvmC and cypX*) and transport (*yvmA*). Our work identifies members of the transition state regulators, ScoC and AbrB, as direct negative regulators of pulcherrimin biosynthesis by inhibiting expression of the biosynthesis gene *yvmC*. The location of the protection areas in the *yvmC* promoter suggest all transcription factors act to inhibit binding and/or progression of RNAP. Nutrient deprivation relieves the repressive effect of the transition state regulators ScoC and AbrB. ROS relieves PchR-mediated repression of the pulcherrimin biosynthesis and transporter operons in *B. licheniformis*, however this has yet to be experimentally verified in *B. subtilis* [46].

[7]. While hundreds of genes were found to be differentially regulated in the *ΔscoC* background, expression of genes involved in pulcherrimin biosynthesis were not identified as significantly different from the WT background [7]. This could be due to the rich, complex media used during the experiment, as *B. subtilis* grown in media with certain amino acids present have lower levels of *scoC* expression compared to media without amino acids [10,31].

*In vivo*, AbrB has been shown to bind to the promoter region for *yvmC* using a ChIP-seq approach [11]. The 53 base-pair binding site partially overlaps with the 90 base-pair region protected by AbrB *in vitro* (Fig 7A). It is possible that our larger protection area represents AbrB binding in the absence of other proteins, such as PchR and ScoC. Perhaps, when present together, lower affinity AbrB binding is competed away, resulting in a smaller binding region when compared to AbrB alone *in vitro*. Interestingly, an AbrB binding site was also found in the upstream promoter region of the co-transcribed genes *pchR-yvmA* [11]. However, transcriptomic data found little change in gene expression of *pchR* and *yvmA* in an *abrB* deletion background [11]. How ScoC contributes to inhibiting expression of the *pchR-yvmA* operon is a current topic of research.

AbrB was also identified as a direct regulator of pulcherrimin biosynthesis in *B. licheniformis* [9,11,15]. Interestingly, the *abrB* deletion had a larger effect on maximum pulcherrimin production in *B. licheniformis* than in *B. subtilis*, indicating that despite similar regulatory components, their effects appear species specific [15]. Further, the homolog for *pchR* is not located adjacent to the pulcherrimin biosynthesis gene cassette like *B. subtilis* (Fig 1B). A

neighboring MarR-family transcription factor YvnA was also identified as a regulator of pulcherrimin biosynthesis in *B. licheniformis*, where it bound directly to the intergenic region between *yvmA* and *yvmC* [15]. One possibility is that the different genetic organizations of the pulcherrimin biosynthetic gene cassette may necessitate alternative forms of regulation, thus explaining the difference between *B. licheniformis* and *B. subtilis*.

In *B. subtilis*, the *pchR* gene is located adjacent to *yvmA* and divergently transcribed from *yvmC-cypX* (Fig 1B) and binds to a consensus motif (PchR-box) upstream of *yvmC* and *pchR*. Our footprinting analysis identified the same region in *yvmC* protected from DNase treatment, validating the predicted motif [13]. MarR family transcription factors commonly bind to small ligands or are covalently modified which alters their DNA binding capabilities [45]. While no small ligand has been identified as an allosteric regulator of PchR activity, He and colleagues have identified ROS as a signal that alleviates DNA binding by PchR in *Bacillus licheniformis* [46]. Specifically, a cysteine residue at position 57 facilitates intermolecular crosslinking in oxidative conditions, decreasing its ability to bind DNA [46]. While this has not been shown in *B. subtilis*, C57 in *B. licheniformis* is structurally conserved in *B. subtilis* (C55), suggesting ROS acts as a signal to induce pulcherrimin production in *B. subtilis* (Fig 7B).

The reason *B. subtilis* and other organisms produce pulcherrimin is a topic of interest. Recent studies found that *B. subtilis* in biofilms produce pulcherrimin as a form of niche protection by creating a zone of iron limitation around the biofilm [12]. Other groups found that pulcherrimin production increased resistance to ROS, likely by decreasing the amount of iron available for Fenton reactions [17,18]. The role of an antioxidant is interesting given that DNA binding ability by PchR in *B. licheniformis* is reduced in oxidative conditions [46]. Consequently, oxidative stress appears to relieve one layer of repression on pulcherrimin production. Our finding that TSRs negatively control pulcherrimin production is notable due to the relationship between stationary phase and ROS tolerance in *B. subtilis*. Indeed, stationary phase cells tend to be more tolerant to hydrogen peroxide than cells in exponential phase independent of prior exposure to ROS [47]. Additionally, production of the cytosolic mini-ferritin MrgA is increased as cells transition into stationary phase in a mechanism independent of TSRs [48]. As a Dps homolog, MrgA is predicted to sequester iron and enzymatically oxidize Fe(II) to the insoluble and less reactive Fe(III) [49]. Thus, control by ROS and TSRs agrees with an antioxidant role for pulcherrimin wherein intracellular and extracellular iron is sequestered as cells enter stationary phase, limiting proliferation of reactive oxygen species.

## Supporting information

**S1 Fig. Iron Supplementation Influences Pulcherrimin Phenotype in WT and isogenic mutants of *Bacillus subtilis*.** 10 μL spots of WT and isogenic mutants on TSS (top) or LB (bottom) supplemented with different concentrations of ferric citrate. The black scale bar represents 5 mm.
(TIF)

**S2 Fig. Cyclo-(L-leucine-L-leucine) Detection from WT and *yvmC::erm* Backgrounds.** Metabolites were extracted from WT (A) and *yvmC::erm* (B) grown in liquid culture and were subject to mass spectrometry analysis for cyclo-(l-leucine-l-leucine), a precursor metabolite for pulcherrimin. The experiment was repeated at least three times with representative data shown. RT (retention time) and S/N (signal to noise ratio) for the peak corresponding to cLL are shown in each panel. The S/N ratio for *yvmC::erm* was under the limit for detection (UD, undetermined).
(TIF)

**S3 Fig. Complementation of *scoC::erm* Restores WT Pulcherrimin Phenotype.** Liquid pulcherrimin measurements from WT, *scoC::erm*, and *scoC::erm lacA::$_p$scoC-scoC* from late stationary phase cultures grown in TSS medium. Bars represent the mean $A_{410}$ from five independent replicates.
(TIF)

**S4 Fig. Absorbance at 600 nm taking during pulcherrimin sampling for Fig 3.** Circles represent average Abs. 600 nm and error bars represent standard deviation.
(TIF)

**S5 Fig. Addition of Heparin Abolishes ScoC-DNA Complexes.** Similar to Fig 5C except heparin was added to the reactions. Fluorescent DNase I footprinting (i) and DFACE analysis with ScoC (ii). In the top panel, red electropherograms represent no protein while blue electropherograms represent reactions with protein.
(TIF)

**S1 Table. Strains Used in this Study.**
(DOCX)

**S2 Table. DNA fragments and Plasmids Used in this Study.**
(DOCX)

**S3 Table. Oligonucleotides Used in this Study.**
(DOCX)

**S1 Data. Numerical data underlying graphs.**
(ZIP)

## Acknowledgments

We would like to thank members of the Simmons lab for helpful discussions during the progression of this work, the Michigan State University Mass Spectrometry and Metabolomics Core for quantitative measurements of intracellular cyclo-(L-leucine-L-leucine), and J.M. Fernandez for constructing the light box used to image agar plates.

## Author Contributions

**Conceptualization:** Nicolas L. Fernandez.

**Data curation:** Nicolas L. Fernandez.

**Formal analysis:** Nicolas L. Fernandez.

**Funding acquisition:** Lyle A. Simmons.

**Investigation:** Nicolas L. Fernandez.

**Methodology:** Nicolas L. Fernandez.

**Project administration:** Lyle A. Simmons.

**Resources:** Lyle A. Simmons.

**Supervision:** Lyle A. Simmons.

**Visualization:** Nicolas L. Fernandez.

**Writing – original draft:** Nicolas L. Fernandez.

**Writing – review & editing:** Nicolas L. Fernandez, Lyle A. Simmons.

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
