## [Decision Letter · Decision Letter 0]

26 Jan 2024

Dear Dr Simmons,

Thank you very much for submitting your Research Article entitled 'Two Distinct Regulatory Systems Control Pulcherrimin Biosynthesis in *Bacillus subtilis*' to PLOS Genetics.

The manuscript was fully evaluated at the editorial level and by independent peer reviewers. The reviewers appreciated the attention to an important topic but identified some concerns that we ask you address in a revised manuscript.

We therefore ask you to modify the manuscript according to the review recommendations. Your revisions should address the specific points made by each reviewer.

Yours sincerely,

Kai Papenfort

Academic Editor

PLOS Genetics

Sean Crosson

Section Editor

PLOS Genetics

Dear Dr Simmons.

Thank you again for submitting your manuscript to PLOS Genetics. Your manuscript has now been evaluated by three referees. As you will see from the comments below, all three referees are positive about your

work and request only few changes to the manuscript. Please make sure to respond to all the comments in your revised manuscript and the rebuttal letter.

Best regards.

Kai Papenfort

Academic Editor

Reviewer's Responses to Questions

**Comments to the Authors:**

Reviewer #1: The authors characterize the roles of two proteins ScoC and AbrB in the repression of pulcherrimin synthesis in B. subtilis. Pulcherrimin is a somewhat unusual compound that sequesters iron extracellularly and is made when iron levels are high, rather than low. They serendipitously notice that a strain lacking the DNA binding protein ScoC overproduces pulcherrimin in liquid culture. They also show that AbrB also represses pulcherrimin genes, as was predicted by bioinformatics analysis but not previously shown. Finally, they make a strain lacking ScoC, AbrB and PhcR and it additively hyperproduces pulcherrimin with the largest contribution made to repression by PhcR. Biochemically, all three proteins bound to the PyvcM promoter, and while AbrB and ScoC binding was much weaker, I think all three proteins bound to the same 59 base pair region within the promoter. All told, the figures and physiological data are extremely nice. The biochemistry I was less confident in, particularly the footprinting as AbrB and ScoC protection seemed rather nonspecific. Finally, I think these results will help us get a better understanding of what pulcherrimin is, what it does, and when/where it does it, but I find myself hard pressed to articulate specifically what it tells us. Finally, the model is appropriately vague, but it feels uncomfortable to have three proteins binding to the same place and have additive effects rather than stepwise effects dominated by proteins in order of highest affinity.

Line 55 and title. Why do these lines say two distinct regulatory systems when three different regulators are discussed (PhcR, ScoC, AbrB)?

Line 106. Clarify “under control”. Are the genes activated or repressed?

Line 467. “positive epistasis”. Do you mean an “additive effect” or “synergy”? Epistasis typically refers to a double mutant having the phenotype of one of the single mutants.

Are the biomass yields the same for each mutant? Just curious as to whether pulcherimin production is dependent on the number of bacteria making it.

Line 496. It is curious that the reporter would have a genetic effect. The following line suggests construction was attempted by transformation into a strain defective in transformation. Have you tried transduction? Or are the authors thinking it has something to do with either a cis or trans factor encoded within the promoter region of yvmC?

Line 522. Something about the term “mutant probe” really threw me off. Mutant usually refers to an in vivo alteration to the sequence. Here a region was deleted from an in vitro target. This may be an overreaction but I really got hung up here because I couldn’t understand how deleting the -10 box would be helpful for in vivo analysis. Mutant may be common usage in biochemistry I’m not sure. Maybe “mutated (or more specifically: deletion) probe”?

Paragraph starting line 521. This paragraph is very long, and covers three different proteins. It could possibly be broken into setup and PhcR (known), with a separate paragraph(s) dedicated to the two new candidates. Or split into EMSA and footprint paragraphs. Whatever the case, it read as too dense and difficult to follow especially for the final section of results where a strong point is needed.

Is it correct that DNA binding of PhcR, AbrB and ScoC were all abolished when using the D59 deletion?

Line 554. Which seemingly redundant pathways are being referred to here? Clarify.

Line 557. Does the triple mutant that overexpressed pulcherrimin have a growth defect, or suffer numerically in competition when grown with the wild type?

Line 584. Clarify “transcriptionally regulated”. Are they activated or repressed?

Line 620. “No pigment was produced”. Implicit in this sentence and the next is that pigmentation (color) is the product of pulcherrimin conjugated to iron suggesting that the pulcherrimin was pre-existent at the time of iron addition so that color change was immediate. Do I have this right?

Figure 1. Do the authors have either a double mutant of abrB ycmC or a quadruple mutant of pchR abrB scoC yvmC? This could control for other siderophores/pigments under AbrB control. I ask because the control was done for scoC but not abrB.

Figure 3A. The gray bars are a wild type lacking the GFP reporter. Are the colored bars only the intensities of the subpopulations that express the reporter? Could the no GFP control be included as a separate panel and then the histograms of both the “on” and “off” subpopulations be shown for the mutants? This way, the bias of expression could be visually represented as well as the intensity maxima. Or perhaps the whole population is shown and the percent off is being calculated as the part that overlaps with the control? Whatever the case, please describe how the percent expressed is being determined.

Reviewer #2: The biofilm-forming and plant-roots associated Gram-positive soil bacterium Bacillus subtilis synthesizes (pulcherriminic acid) and excretes an iron chelating compound (pulcherrimin). It is thought that pulcherrimin serves as an agent that creates a zone of iron limitation around the cell/colony thereby hindering the growth of microorganisms competing for the same ecological niche. Other data associate pulcherrimin production with a role in increasing the resistance levels of the B. subtilis cells against toxic reactive oxygen species. – The genes encoding the enzymes for pulcherriminic acid synthesis cluster with a gene encoding a MarR-type regulator (PchR) and a gene for an export system (YvmA) for pulcherriminic acid (see Fig.6).

In this manuscript, N.L. Fernandez and L.A. Simmons have studied the transcriptional regulation of the operon encoding the pulcherriminic acid synthesis cluster and their data show that it is regulated in an intricate and intertwined fashion by two – more general acting - transition state regulators (AbrB; ScoC) and the more system-specific PchR repressor.

The data presented by the authors are novel and further substantially our understanding of the factors that control an ecophysiological important compound, pulcherrimin. While other studies have already addressed to some extend the genetic control mechanisms for production of pulcherriminic acid, the data presented in this manuscript go well beyond the already known facts. They also correlate the synthesis of pulcherriminic acid/ pulcherrimin with the growth phase of the cells. Hence, in this study genetic control is integrated with a view on physiological relevant parameters for growth (exponential phase versus stationary phase).

The manuscript is written in a very clear and concise fashion, thus aiding the readers in their understanding why a particular experiment was conducted and what the outcome of the experiment was. The experimental set-up of the study is well-controlled, and the description of the Methods section is excellent.

Consequently, I have no major criticism of the data or the interpretation presented in the manuscript but I have some suggestions that the authors might want to consider to improve the presentation of the manuscript.

1. While the authors clearly spell out the current concepts of the physiological roles of pulcherriminic acid/ pulcherrimin, they do this only in the “Discussion” (lines 627-631). I suggest that this important topic should be taken up already in the “Introduction” to the readers can more readily grasp the physiological context of the genetic studies conducted by the authors.

2. In an unusual fashion, the authors refer to Fig. 6 in the “Introduction” (line 103) before they refer to Fig 1. The convention is that figures should be numbered according to the ranking they are referred to in the main text. – Having said that, I find Fig. 6 very important for aiding the reader to comprehend the genetics and physiology of the entire presented story. So, I am suggesting the Fig. 6A,B will become Fig 1. – The current Fig. 6B can also be improved somewhat by being more precise in indicating the positions of the DNA-segments recognized by PchR, ScoC and AbrB, as deduced by the DNA-foot-printing analysis conducted by the authors.

3. In this vain, it would be helpful that have a Fig (perhaps for the Suppl) that shows the actual DNA-sequence of the regulatory region (represented by the extended bar in Fig. 6B) and highlights the actual regions protected in the foot-printing analysis of the various tested transcription factors.

4. The authors have focused their analysis on the operon (yvmC-cypX) encoding the enzymes mediating pulcherriminic acid synthesis (this is fine!). However, one wonders what is known about the gene-cluster (operon?) encoding the pchR regulator and the YvmA pulcherriminic acid exporter. Is pchR autoregulated by its gene product as is observed for many other genes encoding MarR-type regulators)? Any information available for a potential regulatory role of AbrB and ScoC for the pchR-yvmA gene cluster). – This issue should be addressed in the Discussion, as one would expect that the transcription of the gene for the export system would be similarly regulated as those encoding the biosynthetic enzymes.

Reviewer #3: This is a solid and convincing study on the impact of 3 regulators, AbrB, ScoC, and PchR on the expression of genes responsible for pulcherimic acid biosynthetsis in B. subtilis. The finding that ScoC is involved is the only truly novel aspect of this regulation, since both PchR and AbrB were already known or indicated to be involved. Nevertheless, this thorough analysis provides a first comprehensive and convincing treatise on regulation of the chelator. But it remains are rather specific affair for a rather specific audience. The experiments are well-designed, the conclusions justified, the Figures suitable to get the major points across.

Specific points:

Line 400: Since this Fig. is cited here for the first time, it should be Fig. 2, right?! Please change order of Figures accordingly.

Line 496: Any ideas as to why the double mutant, when combined with the reporter, cannot be generated? After all, the double mutant itsel

---

## [Decision Letter · Decision Letter 1]

27 Apr 2024

Dear Dr Simmons,

Thank you very much for submitting your Research Article entitled 'Two Distinct Regulatory Systems Control Pulcherrimin Biosynthesis in *Bacillus subtilis*' to PLOS Genetics.

The manuscript was fully evaluated at the editorial level and by independent peer reviewers. The reviewers appreciated the attention to an important topic but identified some concerns that we ask you address in a revised manuscript.

We therefore ask you to modify the manuscript according to the review recommendations. Your revisions should address the specific points made by each reviewer.

Yours sincerely,

Kai Papenfort

Academic Editor

PLOS Genetics

Sean Crosson

Section Editor

PLOS Genetics

Dear Dr. Simmons.

I am glad to say that the reviewers have commented enthusiastically on your revised paper and your manuscript can now be considered as ‘accepted in principle’. However, before I formally accept the paper, please take a close look at the comment provided by reviewer #1. He raises an important point about the the definition of epistasis that I think should be considered.

Best wishes and congratulations on a very nice manuscript,

Kai Papenfort

(Academic Editor)

Reviewer's Responses to Questions

**Comments to the Authors:**

Reviewer #1: The authors have addressed my concerns. Thank you for the nice paper.

That said, I want to reiterate that the definition of epistasis they are using does not conform to the classical definition of epistasis in bacterial genetics. Epistasis occurs in a double mutant when one mutation masks the phenotype of another. I think the authors definition may come from/be influenced by mis-terminology from field of evolutionary biology, a field which frequently misnames things to suit their purposes and erodes concepts I personally care about. With the evolutionary definition every double mutant can be called epistatic as something always happens when you put two mutations in the same strain, either a positive, negative or neutral effect and is therefore effectively meaningless without the pos/neg qualifier (which stands on its own), whereas the classical definition tells you the order gene products act in a particular pathway. Finally, there are already the terms additive or synergistic effects to describe such outcomes without diluting/erasing/confusing the usefulness of the term epistasis. The authors can of course do/think as they like, but as they are influential leaders in the field, I would prefer they did not reinforce a poor/misguided and ultimately uninformative redefinition. This is important to me, and I ask them to reconsider.

Dan Kearns

Reviewer #2: In this manuscript, the authors provide a solid revision of their original manuscript that addresses the genetic control of the synthesis and export of the iron-chelating compound pulcherrimin in Bacillus subtilis. – In their revision of the manuscript, the authors have taken all comments/questions by the three reviewers into account and carefully addressed them in a comprehensive fashion. I have no further suggestions or questions. – Overall, this is a very nice manuscript, which significantly moves the field forward.

Reviewer #3: Thank you very much for this careful revision.

**Have all data underlying the figures and results presented in the manuscript been provided?**

Reviewer #1: Yes

Reviewer #2: Yes

Reviewer #3: Yes

PLOS authors have the option to publish the peer review history of their article (what does this mean?). If published, this will include your full peer review and any attached files.

Reviewer #1: **Yes: **Daniel Kearns

Reviewer #2: No

Reviewer #3: No

---

## [Editor Report · Decision Letter 2]

3 May 2024

Dear Dr Simmons,

We are pleased to inform you that your manuscript entitled "Two Distinct Regulatory Systems Control Pulcherrimin Biosynthesis in *Bacillus subtilis*" has been editorially accepted for publication in PLOS Genetics. Congratulations!

Yours sincerely,

Kai Papenfort

Academic Editor

PLOS Genetics

Sean Crosson

Section Editor

PLOS Genetics

Comments from the reviewers (if applicable):

Dear Dr Simmons.

Thanks for taking care of this. I have now accepted the paper for publication.

Congratulations again on a very nice manuscript.

Best wishes,

Kai Papenfort

**Data Deposition**

http://datadryad.org/submit?journalID=pgenetics&manu=PGENETICS-D-24-00043R2

**Press Queries**

---

## [Editor Report · Acceptance letter]

11 May 2024

PGENETICS-D-24-00043R2 

Two Distinct Regulatory Systems Control Pulcherrimin Biosynthesis in *Bacillus subtilis*

Dear Dr Simmons, 

We are pleased to inform you that your manuscript entitled "Two Distinct Regulatory Systems Control Pulcherrimin Biosynthesis in *Bacillus subtilis*" has been formally accepted for publication in PLOS Genetics! Your manuscript is now with our production department and you will be notified of the publication date in due course.

With kind regards,

Zsofia Freund

PLOS Genetics

On behalf of:
